# The Ivermectin Related Compound Moxidectin Can Target Apicomplexan Importin α and Limit Growth of Malarial Parasites

**DOI:** 10.3390/cells14010039

**Published:** 2025-01-02

**Authors:** Sujata B. Walunj, Geetanjali Mishra, Kylie M. Wagstaff, Swati Patankar, David A. Jans

**Affiliations:** 1Nuclear Signaling Laboratory, Monash Biomedicine Discovery Institute, Department of Biochemistry and Molecular Biology, Monash University, Clayton, VIC 3800, Australia; sujatawlnj@gmail.com (S.B.W.); kylie.wagstaff@monash.edu (K.M.W.); 2Molecular Parasitology Laboratory, Department of Biosciences and Bioengineering, IIT Bombay, Powai, Mumbai 400076, India; geetanjali_m0012@iitb.ac.in (G.M.); patankar@iitb.ac.in (S.P.)

**Keywords:** *Plasmodium falciparum*, malaria, *Toxoplasma gondii*, toxoplasmosis, importins, nuclear transport inhibitors

## Abstract

Signal-dependent transport into and out of the nucleus mediated by members of the importin (IMP) superfamily is crucial for eukaryotic function, with inhibitors targeting IMPα being of key interest as anti-infectious agents, including against the apicomplexan *Plasmodium* species and *Toxoplasma gondii*, causative agents of malaria and toxoplasmosis, respectively. We recently showed that the FDA-approved macrocyclic lactone ivermectin, as well as several other different small molecule inhibitors, can specifically bind to and inhibit *P. falciparum* and *T. gondii* IMPα functions, as well as limit parasite growth. Here we focus on the FDA-approved antiparasitic moxidectin, a structural analogue of ivermectin, for its IMPα-targeting and anti-apicomplexan properties for the first time. We use circular dichroism and intrinsic tryptophan fluorescence measurements to show that moxidectin can bind directly to apicomplexan IMPαs, thereby inhibiting their key binding functions at low μM concentrations, as well as possessing anti-parasitic activity against *P. falciparum* in culture. The results imply a class effect in terms of IMPα’s ability to be targeted by macrocyclic lactone compounds. Importantly, in the face of rising global emergence of resistance to approved anti-parasitic agents, the findings highlight the potential of moxidectin and possibly other macrocyclic lactone compounds as antimalarial agents.

## 1. Introduction

Signal-dependent transport of molecules into and out of the cell nucleus facilitated by multiple distinct members of the importin (IMP) superfamily of α and β types is central to eukaryotic cell function [1]. The best understood nuclear import pathway is that mediated by a heterodimer of IMPα and IMPβ1, where IMPα binds specifically to IMPβ1 and to the nuclear localisation signal (NLS) of cargo proteins [2,3,4,5]. IMPβ1 within the complex then mediates passage through the nuclear pore into the nucleus and subsequent release in the nucleus via distinct binding interactions with various other components of the nuclear transport machinery [1,2,3,4,5].

Dysregulation of nucleocytoplasmic transport can severely impact cell function [1,6,7,8], with the inhibition of transport through the use of small molecules a promising therapeutic strategy for cancer and infectious diseases [1,4,9]. Various inhibitors of IMPα such as the macrocyclic lactone ivermectin [10] have been a focus of interest as antiviral agents [1,4,9,11,12,13,14,15,16], and more recently as agents to combat the apicomplexan *Plasmodium* species and *Toxoplasma gondii* [17,18], the causative agents of malaria and toxoplasmosis, respectively [19,20,21,22,23,24].

With 249 million cases and 608,000 deaths in 2022 worldwide alone, malaria is a global concern (WHO Malaria Report, 2023), with growing resistance to artemisinin-based combination therapies highlighting the need for new antiparasitic approaches [19,20,21]. Toxoplasmosis, caused by *T. gondii*, potentially afflicts one-third of humans asymptomatically, with immunocompromised individuals and children facing severe complications; the limitations of current therapies for toxoplasmosis, including drug resistance, emphasize the compelling need to develop new therapeutic strategies [22,23,24,25].

We recently showed that distinct inhibitors targeting IMPα can limit the growth of *P. falciparum* and *T. gondii* in culture [17,18], validating IMPα from these parasites as viable therapeutic targets [26,27,28,29]. These inhibitors included the US Food and Drug Administration (FDA)-approved anti-helminthic ivermectin [10,30], which is active against a range of different viruses, including SARS-CoV2 [9,12,13,14,15,16,31,32,33], and has been clinically investigated in an anti-malarial context [34,35]. Of a number of structural homologues of ivermectin that belong to the macrocyclic lactone class of compounds and possess anti-parasitic properties, moxidectin is also FDA-approved for human use [36,37]. Both moxidectin and ivermectin possess the ability to target glutamate-gated chloride channels [37], with moxidectin showing longer half-life/sustained release into the bloodstream that may be part of the basis of its more potent and prolonged anti-parasitic effect compared to ivermectin [38].

Although moxidectin has demonstrated efficacy against various parasites in animals [37,38,39,40], its specific activity against apicomplexan parasites, or apicomplexan IMPα, has not previously been examined. The present study addresses this directly for the first time, comparing the effects of moxidectin with those of ivermectin on IMPα from *P. falciparum* and *T. gondii*, as well as its anti-parasitic activity. Moxidectin shows a robust ability to interact with and inhibit the functions of apicomplexan IMPα, as well as limit malarialparasite growth in culture. The results imply a class effect of macrocyclic lactones in this context, opening up the intriguing possibility of using these compounds more widely in the context of malaria and toxoplasmosis.

## 2. Materials and Methods

### 2.1. Protein Expression and Purification

*P. falciparum* IMPα (PfIMPα, PlasmoDB ID: Pf3D7_0812400), *T. gondii* IMPα (TgIMPα, ToxoDB ID: TGGT1_252290), *Mus musculus* IMPα (Kpna2/Rch1/pendulin/PTAC58 NCBI ID: 16647) abbreviated here to MmIMPα; MmIMPα deleted for the IMPβ1-binding domain (IBB) (MmΔIBBIMPα); and *M. musculus* IMPβ1 (MmIMPβ1, NCBI ID: 16211) GST fusion proteins were all expressed and purified as previously [14,16,17,18]. The His-tagged green fluorescent protein (GFP) fusion proteins T-ag-NLS-GFP and TGS1-NLS-GFP, as well as PfIMPα and TgIMPα proteins, were all purified using nickel-affinity chromatography as described [14,16,17,18]. GST-fusion proteins were biotinylated using the Sulfo-NHS-Biotin reagent (Pierce, Rockford, IL, USA) as previously [14,17].

### 2.2. Compounds

Moxidectin and ivermectin were from Sigma-Aldrich (St. Louis, MO, USA); 10 mM stock solutions were prepared in 100% DMSO.

### 2.3. AlphaScreen

IC_50_ analysis was performed in quadruplicate using an AlphaScreen binding assay as previously [16,17,18], using bacterially expressed glutathione S-transferase (GST)-C-terminally tagged PfIMPα and TgIMPα proteins [17], as well as SV40 T-ag-NLS-GFP fusion protein and MmIMPβ1 [11,41]. As previously, nickel-acceptor and streptavidin-donor beads were used [17,18]. The AlphaScreen signal was measured on a Perkin Elmer plate reader, and IC_50_ plots drawn using GraphPad Prism 9.0.2 (San Diego, CA, USA).

### 2.4. Circular Dichroism (CD) Spectroscopy

CD spectroscopy experiments were carried out to assess direct binding of moxidectin and ivermectin to different IMPαs as described [17,18]. CD spectra at 90.15 mg/mL protein concentration were recorded from 200–260 nm using a Jasco CD spectrometer (Jasco, Easton, MD, USA) in the absence and presence of 30 or 80 μM ivermectin or moxidectin, with percentage α-helix content estimated from the ellipticity at 222 nm as described [16,42] using the CD Multivariate secondary structure estimation analysis program.

### 2.5. Intrinsic Tryptophan Fluorescence Assays

Tryptophan fluorescence spectroscopy experiments were performed using a JASCO Fluorescence Spectrophotometer (JASCO, Hachioji, Tokyo, Japan) (0.5 mL quartz cuvette), with a fixed excitation wavelength of 295 nm and emission spectra collected between 310 and 400 nm with a slit width of 5 nm at 25 °C. Increasing concentrations of compounds in DMSO were incubated with the IMPαs (1 µM) in PBS for 5 min at room temperature prior to measurement. Data from three independent experiments were analysed using GraphPad Prism 10 (San Diego, CA, USA). Non-linear regression analysis was used to fit the one-site binding curves using the equation: y = Bmax * x/(Kd + x), where Bmax represents the maximum binding capacity and Kd is the dissociation constant, reflecting the concentration of ligand at which half-maximal binding is achieved.

### 2.6. P. falciparum Culture

The *P. falciparum* 3D7 strain was cultured and maintained according to the standard procedures [43] with minor modifications. Cultures were maintained in RPMI 1640 (GibcoTM, Waltham, MA, USA) supplemented with 0.5% Albumax (GibcoTM), 50 mg/L hypoxanthine (Sigma-Aldrich), 2 gm/L D-glucose (Sigma-Aldrich), 2 gm/L sodium bicarbonate (Sigma-Aldrich), and 56 mg/L of gentamicin (Abbott, Chicago, IL, USA). Blood for parasite cultures was procured from volunteer blood donors with approval from the Institute Ethical Committee.

### 2.7. Drug Susceptibility Assays for P. falciparum Using HRP2 (Histidine-Rich Protein 2) ELISA

Samples from the continuous culture of *P. falciparum* were synchronised to obtain predominant rings of 0.25% parasitemia at 3% haematocrit. Compounds were made at a stock concentration of 10 mM in 100% DMSO, with the FDA-approved antimalarial dihydroartemisinin (a kind gift from IPCA Laboratories, Mumbai, India) as a control. Compound dilutions were made in RPMI 1640 to obtain the desired final test concentrations for the IC_50_ analysis.

The HRP2-sandwich ELISA assay was performed according to the standard protocol [16,17,44,45]. Briefly, the PfHRP2 protein concentration in the culture wells, indicative of the level of parasitemia, is measured using a sandwich ELISA technique. Serial 2-fold dilutions of compounds (25 µL/well) were manually dispensed into standard 96-well microculture plates (Eppendorf). Subsequently, 200 µL of *P. falciparum* culture at 0.25% parasitemia (3% haematocrit) was added to each well. The plates were then incubated for 72 h at 37 °C in 5% CO_2_ in a humidified incubator. Plates were then subjected to freeze–thaw to achieve complete haemolysis. PfHRP2 levels in control and test samples were measured using a horseradish peroxidase enzyme-linked immunosorbent assay at 450 nm. GraphPad Prism 9.2.0 was used to perform non-linear regression analysis to determine the IC_50_ values [16,17].

### 2.8. T. gondii Tachyzoite Culture and Growth Inhibition Assay

A *T. gondii* RH strain expressing a luciferase reporter (RH-Fluc) [16] was used to assess the effect of the inhibitors on growth as described. *T. gondii* tachyzoites were maintained and cultured at 37 °C in 5% CO_2_ in a humidified incubator using primary human foreskin fibroblasts (HFF, ATCC) maintained in Dulbecco’s modified Eagle medium (DMEM) (GibcoTM, Waltham, MA, USA) supplemented with 3.7 g/L sodium bicarbonate and 2.38 g/L HEPES, 10% Cosmic Calf serum (HycloneTM, Logan, UT, USA) and 20 mg/L gentamicin [46,47]. Growth inhibition was monitored by luminescence [16], where 100 μL of culture medium without or with inhibitors, followed by 100 µL of DMEM containing 5000 parasites, were added to confluent monolayers of HFF cells in 96-well culture dishes (Eppendorf, Hamburg, Germany). After 48 h, 150 µL of the culture medium was discarded from each well, and 10 µL lysis buffer was added to lyse the parasites, followed by 50 μL of 2× luciferase assay reagent (Promega, Madison, WI, USA). Luminescence was then measured directly for 10 s using a Varioskan™ LUX multimode microplate reader (Thermo Fisher Scientific, Waltham, MA, USA).

### 2.9. MTT Assay for Host Cell Cytotoxicity

The MTT assay was used to measure the cytotoxicity of the small molecules against the HFF cells as described [48]. Briefly, freshly confluent HFF cells were incubated for 48 h in a 96-well culture plate at 37 °C in a humidified incubator (5% CO_2_ atmosphere) in the presence of increasing concentrations of compounds (total volume 200 uL). A 10 μL quantity of MTT reagent (Sigma Aldrich) prepared in complete DMEM medium was then added to each well, and incubation continued for a further 3 h. 150 μL of the medium containing the MTT reagent was then removed from each well and replaced with 150 μL DMSO. Plates were then incubated room temperature in the dark for 20 min, after which absorbance for each well was measured at 570 nm. Data were analysed in GraphPad Prism 9 software.

### 2.10. Statistical Analysis

Non-linear regression analysis was performed in GraphPad Prism 9.2.0 (San Diego, CA, USA) to fit four parameter dose–response curves using the formula: y = a + ((b − a)/(1 + 10^(Log(c) − x)^ × d)), where a and b are the minimum and maximum asymptotes respectively, c is the half-maximal inhibitory concentration value (IC_50_), and d is the slope at the steepest part of the curve (the Hill slope).

## 3. Results

### 3.1. Moxidectin Can Block NLS Recognition by Apicomplexan IMPαs

Structurally related to ivermectin (see Figure 1a), the anti-parasitic agent moxidectin was approved for the treatment/prevention of river blindness (onchocerciasis) in humans by the FDA in 2018 [36,37,38]. It appears to have a longer half-life/higher efficacy against *Onchocerca volvulus* than ivermectin in animals and humans [38,49,50]. We recently showed that as for mammalian nuclear transport systems [11,12,15,16], ivermectin can block the interaction of apicomplexan IMPα proteins with NLSs as well as IMPβ1 [17]. Here, we set out for the first time to test whether moxidectin had comparable activity, performing IC_50_ analysis using an established AlphaScreen binding assay [16,17].

In contrast to mammalian systems where there are multiple different IMPα forms [2,3,4], *P. falciparum* and *T. gondii* both retain a single, essential form of IMPα, the amino acid sequences of which are 63 and 59% similar (43 and 41% identical), respectively, to that of MmIMPα (Kpna1/Rch1—see Section 2.1) used here. As previously [16,17,18], we first analysed the MmΔIBBIMPα (truncated) form of IMPα, which is not autoinhibited and hence can bind NLSs with high affinity in the absence of MmIMPβ1 [51,52]. We found that ivermectin inhibited MmΔIBBIMPα recognition of the well-characterised simian virus SV40 large tumour antigen (T-ag) in the context of a bacterially expressed GFP fusion protein (T-ag-NLS-GFP) at low μM concentration (Figure 1b bottom left; IC_50_ value of c. 6 μM—see Table 1), consistent with previous studies [11,16,18,41]. Strikingly, moxidectin showed similar inhibitory activity (Figure 1b bottom right) with an IC_50_ of 3–4 μM (Table 1). As previously [18], ivermectin inhibited NLS binding by PfIMPα (TGS1-NLS-GFP fusion protein) and TgIMPα (SV40:T-ag-NLS-GFP fusion protein) at low μM concentration (Figure 1b, top left and middle left panels), with IC_50_ values of c. 5 µM (Table 1). Moxidectin showed very similar ability to inhibit both apicomplexan IMPαs (Figure 1b top right/middle right panels), with comparable IC_50_ values (4–5 µM, Table 1). Clearly, moxidectin strongly resembled ivermectin in being able to inhibit NLS binding by IMPαs, including those from apicomplexans, at low µM concentrations.

Apicomplexan IMPβ1 has not been characterized as yet, but previous studies [16,18] show that apicomplexan IMPαs are able to interact with mammalian IMPβ1 with high affinity, and that ivermectin is able to inhibit this interaction through its ability to bind to IMPα to alter structure/conformation. Accordingly, we used the AlphaScreen binding assay to assess the ability of moxidectin to inhibit binding of IMPα from *Mus musculus*, *P. falciparum* and *T. gondii* to mammalian IMPβ1. Binding of full-length MmIMPα to MmIMPβ1 was inhibited by moxidectin with an IC_50_ of c. 4 µM, very similar to that of ivermectin (Figure 2, Table 1). Strikingly, it also inhibited binding of PfIMPα and TgIMPα to MmIMPβ1 with IC_50_ values of c. 3 µM, very similar to those of ivermectin (IC_50_ values of 2–7 µM; Figure 2, Table 1). Clearly, moxidectin resembles ivermectin in being able to inhibit IMPβ1 recognition by mammalian and apicomplexan IMPαs at low µM concentrations.

### 3.2. Moxidectin Can Bind Directly to Apicomplexan IMPα Proteins

We employed far-UV CD spectroscopy, as previously [14,18], to assess the potential of moxidectin to bind directly to IMPα proteins. The CD spectra of PfIMPα, TgIMPα, and MmIMPα all showed double minima at 208 and 222 nm (Figure 3a), consistent with the predominantly α-helical structure of IMPα (see [51]). Quantitative estimation indicated that apicomplexan IMPαs were c. 60% α-helical, compared to c. 70% for MmIMPα (Figure 3b). The CD spectra of the proteins in the presence of increasing concentrations of ivermectin and moxidectin revealed a concentration-dependent reduction in α-helicity (Figure 3a,b), indicating that, like ivermectin, moxidectin appears to be able to bind to the IMPαs directly to perturb structure.

We built on these results using an intrinsic tryptophan fluorescence assay. Tryptophan residues within proteins have an intrinsic fluorescence, a change in the microenvironment of which, through ligand binding or other interactions by the protein, can alter the intrinsic fluorescence intensity [53]. We recorded tryptophan fluorescence emission spectra for the IMPαs, all of which showed a fluorescence maximum at 340 nm.

The addition of increasing concentrations of ivermectin or moxidectin resulted in a concentration-dependent decrease in fluorescence intensity at 340 nm (Figure 4a), enabling estimation of dissociation constants (Figure 4b). Moxidectin and ivermectin showed low μM IC_50_ values for mammalian IMPα (3 µM and 4 µM) and PfIMPα (6.4 µM and 7.8 µM). In contrast, IC_50_ values of moxidectin and ivermectin were higher for TgIMPα (24 µM and 19 µM). As NLS binding sites in IMPα have conserved tryptophan residues, these results suggest that moxidectin and ivermectin bind to conserved tryptophan residues on apicomplexan IMPα proteins and impact their conformation; the significance of the possibility that TgIMPα may be slightly less sensitive in this respect is not clear.

**Figure 4 cells-14-00039-f004:**
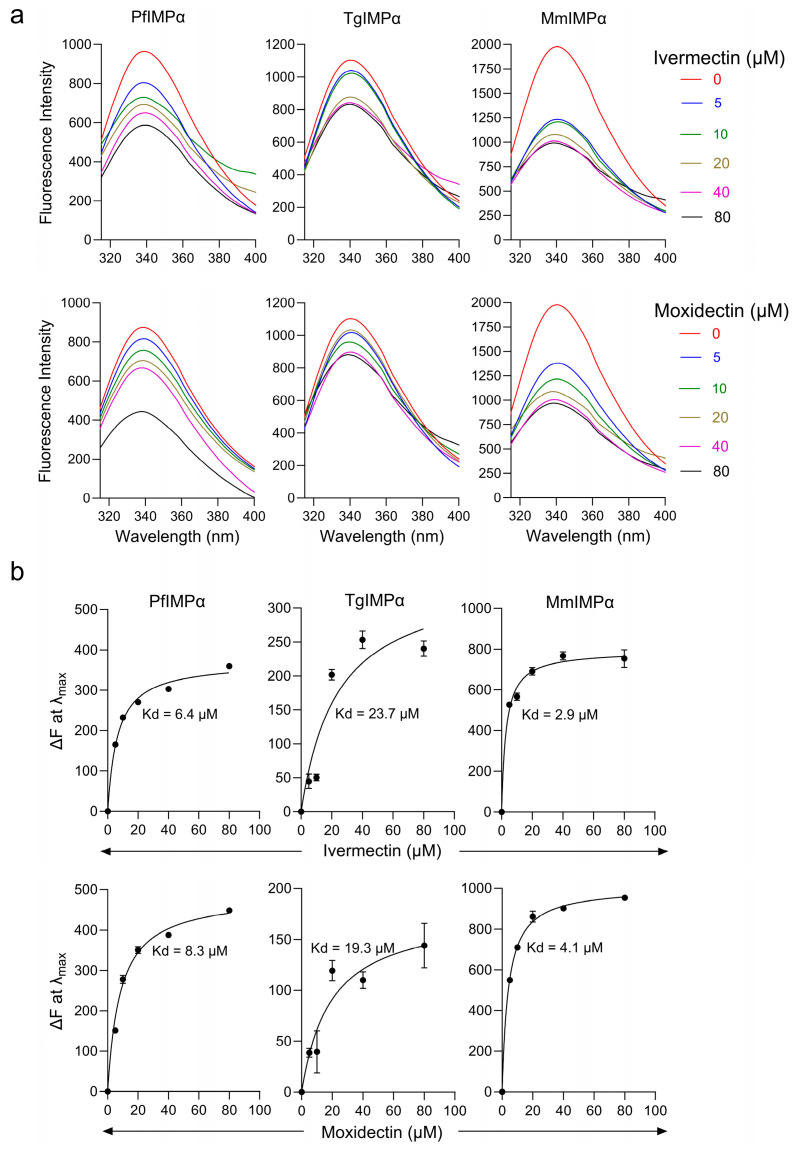
Intrinsic tryptophan fluorescence assay to study the interaction between moxidectin and ivermectin with PfIMPα, TgIMPα, and MmIMPα. (**a**) Intrinsic tryptophan fluorescence spectra of PfIMPα, TgIMPα, and MmIMPα were collected (excitation wavelength: 295 nm and emission from 315 nm to 400 nm) in the presence or absence of moxidectin and ivermectin in the indicated concentration ranges. (**b**). Changes in fluorescence intensity at 340 nm (λmax) from (**a**) with increasing concentrations of compound were plotted using GraphPad prism for a single typical set of measurements, representative of a series of 3 independent experiments (see Table 2).

### 3.3. Like Ivermectin, Moxidectin Can Limit Proliferation of P. falciparum

Since moxidectin appears to bind to PfIMPα and thereby inhibit NLS- and IMPβ1-binding, we carried out drug susceptibility assays for moxidectin and ivermectin against *P. falciparum* parasites using an established ELISA technique [16,17,44,45], with the clinically prescribed drug dihydroartemisinin (DHA) used as a positive control. In similar fashion to ivermectin, moxidectin showed an IC_50_ value of 1 μM against *P. falciparum* asexual stages in culture (Figure 5, Table 3), supporting the idea that moxidectin, like ivermectin, has antimalarial activity.

We also tested the effect of moxidectin on the growth of *T. gondii* tachyzoites constitutively expressing luciferase in HFF host cells [54]. IC_50_ analysis indicated values in the 30–35 µM range, comparable to that for ivermectin (see Table 4). Analysis of the potential cytotoxic effects of the compounds on the HFF cells in the infectious system using the MTT assay, however, revealed that both ivermectin and moxidectin have CC_50_ values for HFF cells of c. 10 µM (Table 4). Clearly, it cannot be excluded that the observed inhibitory effects of ivermectin/moxidectin on *T. gondii* tachyzoites in the HFF cell infectious system are attributable to cytotoxic effects on the host cells.

## 4. Discussion

This study is the first to show that the macrocyclic lactone moxidectin shows promise as an inhibitor of IMPα, and accordingly has potential as an anti-parasitic agent. Through its chemical scaffold shared with ivermectin (see Figure 1a), moxidectin exhibits similar inhibitory effects on mammalian and apicomplexan IMPα proteins. Both compounds effectively inhibit NLS recognition and binding to IMPβ1 with IC_50_ values of 3–7 μM (Table 1; Figure 1 and Figure 2); based on the fluorescence quenching and structural perturbations observed in the tryptophan fluorescence and CD spectroscopy assays here (Figure 3 and Figure 4), conserved tryptophan residues within the IMPαs likely play a critical role in these interactions. Consistent with this, moxidectin’s activity against the asexual stages of *P. falciparum* was confirmed in culture (IC_50_ of 1 μM; Figure 5). Thus, taken together our novel findings suggest that moxidectin, like ivermectin, targets/binds to conserved structural motifs in IMPα, disrupting their conformation and inhibiting key protein–protein interactions; this is likely the basis of its antimalarial activity. As it is approved by the FDA to treat helminthic infections, moxidectin’s pharmacokinetic properties/safety are well documented in human studies (see [36,40,49,50]), but of course, rigorous testing of the antimalarial action of moxidectin in clinical studies, initially along the lines of those already carried out for ivermectin [34,35], will be required in the future to assess properly its true therapeutic potential against apicomplexan disease.

As mentioned, moxidectin and ivermectin share key chemical scaffolds (Figure 1a) that enable them to target glutamate-gated chloride channels in helminths/insects [54,55,56,57,58]. Based on the results here using CD and intrinsic tryptophan fluorescence measurements, as well as previous studies [16], it would appear that moxidectin/ivermectin’s shared chemical scaffolds enable them to bind directly to specific regions of IMPα, thereby impacting key protein–protein interactions central to the nuclear transport process. Understanding the precise binding site of these compounds on apicomplexan and mammalian IMPα is crucial for the development of agents that selectively target apicomplexan without affecting mammalian host IMPα. This understanding holds the potential to pave the way for the development of a novel class of antimalarials that are selective and efficacious.

A key implication of this study is that other macrocyclic lactones related to ivermectin and moxidectin (avermectins, milbemycins etc.) may conceivably have similar properties. Although moxidectin is structurally related to ivermectin, it has distinct features (see Figure 1a and legend), which appear to be the basis of its improved stability and efficacy as an anti-parasitic compared to ivermectin in mammalian systems [38,49]. It is an intriguing possibility that other distinct avermectins/milbemycins, such as selamectin, eprinomectin, abamectin, doramectin, etc., which are currently in veterinary/agricultural use may also prove to have the same general properties. Direct experimentation is of course required to test this possibility, but in the important quest to continue to develop new strategies to combat apicomplexan disease, especially in the context of emerging resistance to existing treatments, this may well prove worthwhile. Moxidectin here showed antimalarial activity at low μM concentrations; combined with its reported longer half-life and sustained release in mammalian systems compared to ivermectin [38,49], it appears to be a more compelling candidate for further exploration in the development of anti-parasitic drugs than ivermectin. Future investigations should focus on optimising moxidectin’s selectivity, exploring its efficacy in vivo, and potentially its specific targets within the parasites.

In summary, the study presented here emphasizes the exciting potential of repurposing existing drugs, such as moxidectin and ivermectin, to address the urgent need for novel and effective treatments against malaria.

## Figures and Tables

**Figure 1 cells-14-00039-f001:**
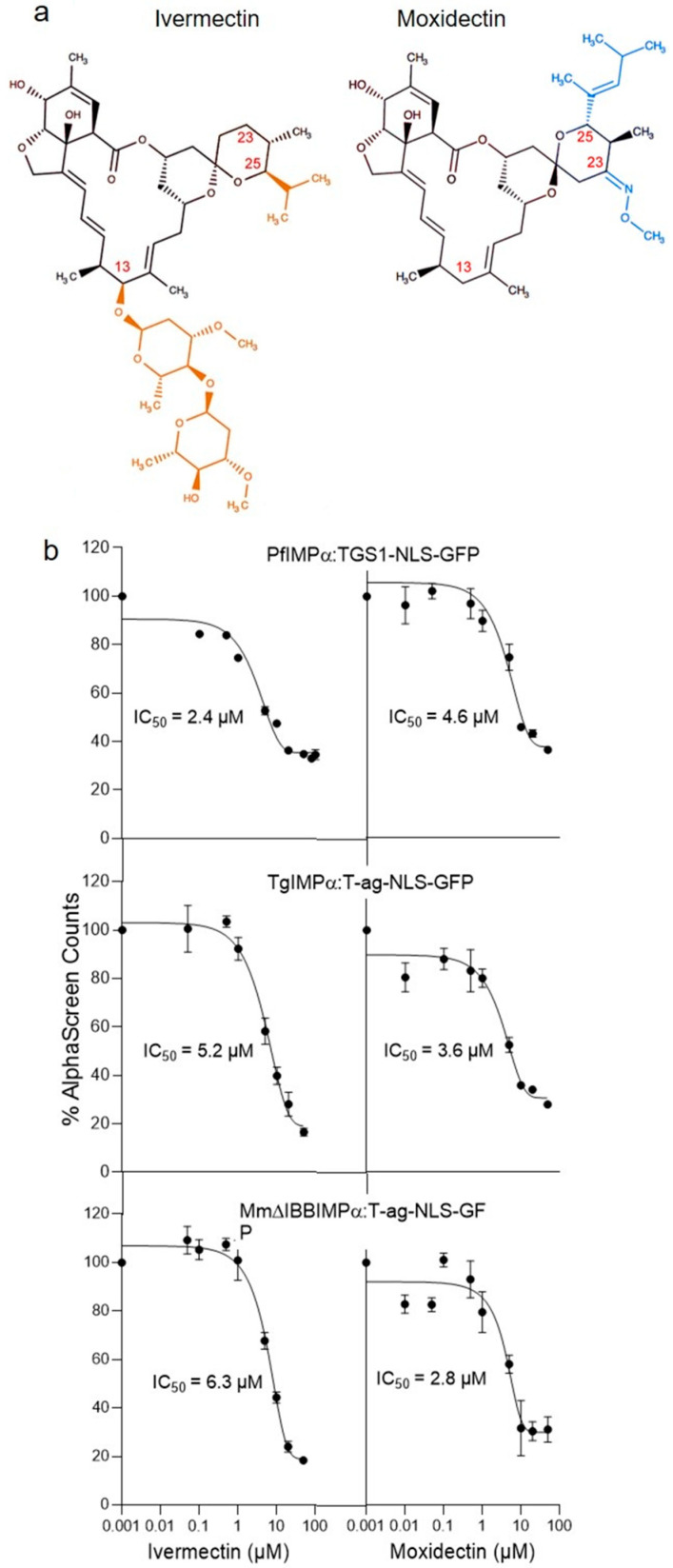
Close similarities of ivermectin and moxidectin in structure and IMPα inhibitory properties. (**a**). Structures of ivermectin and moxidectin are shown, with the shared chemical scaffold in black, and distinct groups highlighted in colour; an example is the C13 residue that is attached to sugar groups in the case of avermectins such as ivermectin, but is protonated in moxidectin and other members of the milbemycin family. (**b**). AlphaScreen technology was used to determine the IC_50_ values for inhibition of the binding of various IMPαs (5 nM) to NLS-containing proteins (30 nM) by ivermectin and moxidectin. Data represent the mean ± SEM for quadruplet wells from a single experiment, from a series of 3 independent experiments (see Table 1 for pooled data).

**Figure 2 cells-14-00039-f002:**
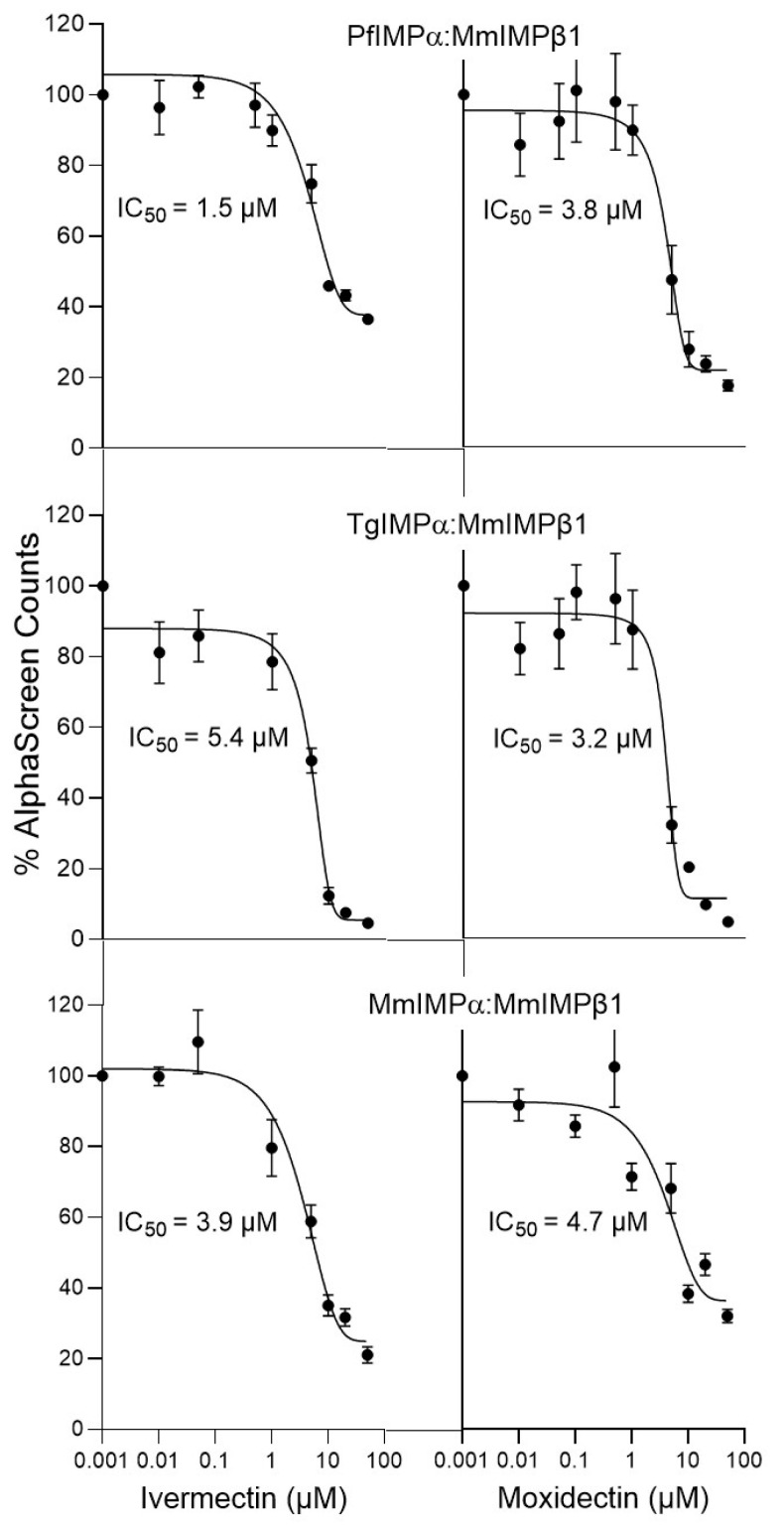
Like ivermectin, moxidectin can inhibit IMPα–IMPβ1 binding. AlphaScreen technology was used to determine the IC_50_ for inhibition by ivermectin and moxidectin of binding of MmIMPβ1 (30 nM) to various IMPαs (30 nM). Data represent the mean ± SEM for quadruplet wells from a single typical experiment, from a series of three independent experiments (see Table 1 for pooled data).

**Figure 3 cells-14-00039-f003:**
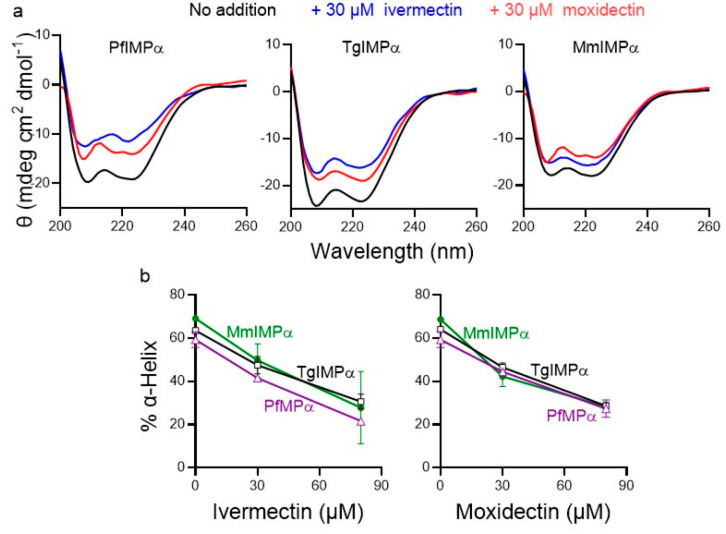
CD spectra for apicomplexan and mammalian IMPαs in the absence and presence of moxidectin and ivermectin. CD spectra were collected for PfIMPα, TgIMPα, and MmIMPα in the absence or presence of 30 or 80 μM ivermectin or moxidectin. (**a**). Spectra are shown from a single experiment, representative of 2 independent experiments for IMPαs without or with 30 μM ivermectin or moxidectin. Note: θ is ellipticity in mdeg cm^2^ dmol^−1^. (**b**). The α-helical content of the respective IMPαs was estimated as previously (see Section 2.4) from spectra as per Figure 3a. Results represent the mean ± SD for 2 independent experiments.

**Figure 5 cells-14-00039-f005:**
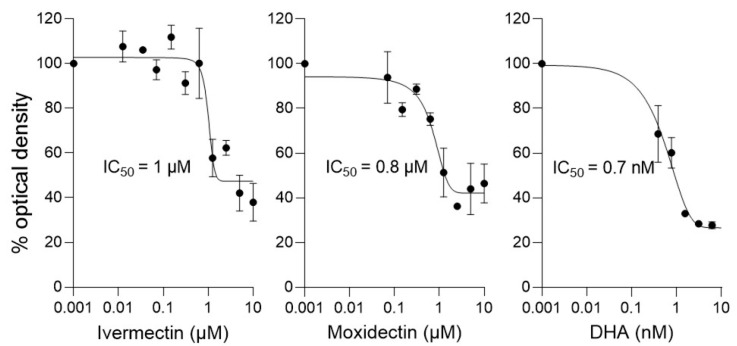
Ivermectin and moxidectin inhibit *P. falciparum* parasites in culture at low µM concentrations. *P. falciparum* cultures (0.25% parasitemia) were treated with increasing concentrations of the indicated compounds for 72 h, after which the HRP2-based sandwich ELISA was used to measure the HRP2 levels, determined by optical density. The results shown are from a single typical experiment performed in duplicate (SD shown), representative of a series of three independent experiments (see Table 3 for pooled data).

**Table 1 cells-14-00039-t001:** Summary of IC_50_ analysis for inhibition of IMPα binding interactions by moxidectin and ivermectin.

	IC_50_ (μM) *	
Binding Interaction	Ivermectin	Moxidectin
PfIMPα:TGS1-NLS-GFP	5.0 ± 1.3	4.8 ± 0.4
TgIMPα:Tag-NLS-GFP	4.9 ± 0.2	3.8 ± 0.8
MmΔIBBIMPα:Tag-NLS-GFP	6.2 ± 0.2	3.6 ± 0.5
PfIMPα:MmIMPβ1	1.9 ± 0.2	3.4 ± 0.4
TgIMPα:MmIMPβ1	6.9 ± 1.2	2.7 ± 0.2
MmIMPα:MmIMPβ1	3.1 ± 0.5	4.0 ± 0.4

* Results represent the mean ± SEM (*n* = 3) for the values measured, as per Figure 1 and Figure 2.

**Table 2 cells-14-00039-t002:** Summary of Kd estimations measured using intrinsic tryptophan fluorescence measurements.

Protein	Kd (μM) *	
	Ivermectin	Moxidectin
PfIMPα	6.4 ± 0.3	7.8 ± 1.2
TgIMPα	24 ± 1.4	19 ± 2.2
MmIMPα	2.9 ± 0.4	4.1 ± 0.2

* Results represent the mean ± SEM (*n* = 3) for the values measured as per Figure 4b.

**Table 3 cells-14-00039-t003:** Summary of the IC_50_ values for inhibition of *P. falciparum* parasites by moxidectin and ivermectin in culture.

Compound	IC_50_ (µM) *
Ivermectin	0.7 ± 0.1
Moxidectin	0.9 ± 0.1
DHA	0.001 ± 0.0005

* Results represent the mean ± SD (*n* = 3) for IC_50_ analysis as per Figure 5.

**Table 4 cells-14-00039-t004:** Summary of IC_50_/CC_50_ values for effects of moxidectin and ivermectin on *T. gondii* parasites in culture.

Compound	IC_50_ (µM) *	CC_50_ (µM) ^+^
Ivermectin	29 ± 0.7	8 ± 1.8
Moxidectin	35 ± 3.1	10 ± 0.8

* Results represent the mean ± SD (*n* = 3). ^+^ Results represent the mean ± SD (*n* = 2).

## Data Availability

The original contributions presented in this study are included in the article. Further inquiries can be directed to the corresponding author.

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
