# Peer review of "The Ivermectin Related Compound Moxidectin Can Target Apicomplexan Importin α and Limit Growth of Malarial Parasites"

_cells, 2025, doi:10.3390/cells14010039_

Round 1

Reviewer 1 Report

Comments and Suggestions for Authors

The article by Sujata B. Walunj et al disclosed that the FDA-approved anti-parasitic drug Moxidectin could inhibit the IMPα in acrocomplex parasites like Plasmodium and Toxoplasma. This finding offers a novel direction for the development of new drugs against malaria and toxoplasmosis, and holds significant scientific and clinical importance. Direct binding of moxicodin to Importin α and its inhibition of protein function were examined in detail through Circular dichroism (CD) spectroscopy and Intrinsic Tryptophan Fluorescence Assays. Howeverit were absent to comprehensively assess its efficacy and safety in practical clinical applications. Given the increasing resistance to current anti-parasitic drugs, the results suggest that moxicodin and other macrolides might become new anti-malaria drugs.

Major comments: Key Experiments Required for Acceptance

1. Although the study disclosed the inhibitory effect of moxicodin on the adventite parasite Importin alpha protein, in vivo data were absent to comprehensively assess its efficacy and safety in practical clinical applications.

2. Further exploration of the metabolic and pharmacokinetic properties of moxigetine in complex biological systems will furnish more comprehensive drug information to facilitate its clinical development.

3. It is proposed to clarify the functional influence of moxicodin based on interaction.

Minor Issues: Editorial and Data Presentation Modifications

Material and Methods:

1. In Line 74-81Kindly provide the gene sequence associated with protein expression.

2. In Line 153-156Please elaborate on the experimental methods and procedures in this section.

Results

1. In Line 173please elaborate on the significance of MmΔIBBIMPα protein construction regarding the topic of this article.

2. Figure 1the chemical structures of the two drugs are presented in the article.  At what specific sites are they targeting the IMP protein?

3. Figure 4What is the purpose of the interaction between moxidectin and  ivermectin with MmIMPα

4. What is the homology between Plasmodium IMP and host-associated protein, and does Moxidectin have an impact on the host?

5. Table 4Ivermectin and Moxetine exhibited high cytotoxicity towards HFF cells.  What was the aim of this experiment, or why were other relevant cytotoxicity experiments not carried out?

6. The picture definition is excessively low. Please make modifications.

Discussion:

1.   In Line 303-307the discussion section is not in line with the research. Please make the necessary revisions.

2.   In Line 318-323“low µM concentration” this description is incorrect. Please amend it.

3.   There is no summary of the main results in the discussion section.  Please add that the discussion should center on the main experimental results.

Author Response

We thank the Reviewer for the positive and probing comments - our responses are below.

The article by Sujata B. Walunj et al disclosed that the FDA-approved anti-parasitic drug Moxidectin could inhibit the IMPα in acrocomplex parasites like Plasmodium and Toxoplasma. This finding offers a novel direction for the development of new drugs against malaria and toxoplasmosis, and holds significant scientific and clinical importance. Direct binding of moxicodin to Importin α and its inhibition of protein function were examined in detail through Circular dichroism (CD) spectroscopy and Intrinsic Tryptophan Fluorescence Assays. However,it were absent to comprehensively assess its efficacy and safety in practical clinical applications.

We thank the Reviewer for the positive comments; the Reviewer can appreciate that efficacy and safety in the clinic was well beyond the scope of the present study; the Reviewer should also be aware that both agents have been approved by the TGA for safe human use. We now point out the importance of the Reviewer’s points here in the revised manuscript (line 318-323) – we thank the Reviewer for his/her important input.

Given the increasing resistance to current anti-parasitic drugs, the results suggest that moxicodin and other macrolides might become new anti-malaria drugs. 

Major comments: Key Experiments Required for Acceptance

  1. Although the study disclosed the inhibitory effect of moxicodin on the adventite parasite Importin alpha protein, in vivo data were absent to comprehensively assess its efficacy and safety in practical clinical applications.

We are a little confused – Figure 5/Table 3 address the specific point of activity against the malarial parasite in an infectious model. As indicated above, analysis of moxidectin’s clinical potential is well beyond the scope of the present manuscript – we expand on the importance of clinical studies in the Discussion of the revised manuscript (line 318-323) – we thank the Reviewer.

  1. Further exploration of the metabolic and pharmacokinetic properties of moxigetine in complex biological systems will furnish more comprehensive drug information to facilitate its clinical development. 

We are a little confused –moxidectin is an FDA approved agent whose metabolic and pharmacokinetic properties are well-documented. We have spelled this out more clearly in the manuscript (lines 59-61; 318-320; Refs 49,50 etc.), and thank the Reviewer for making us clarify this point.

  1. It is proposed to clarify the functional influence of moxicodin based on interaction.

We are a little confused as to what the Reviewer is concerned about here – the present submission shows the effect of moxidectin on importin alpha using a range of different assays, consistent with unfolding of importin alpha (CD analysis; tryptophan fluorescence etc.), thereby inhibiting importin alpha from its functional roles of NLS binding and interaction with importin b1. This underlines the functional influence of moxidectin on importin alpha; we now make this clearer in the manuscript in the second part of the first paragraph of the Discussion (lines 307-318). We thank the Reviewer.

Minor Issues: Editorial and Data Presentation Modifications

Material and Methods:

  1. In Line 74-81,Kindly provide the gene sequence associated with protein expression.

Gene sequence IDs for the key importin proteins are now provided – we thank the Reviewer for pointing out our omission (lines 76-79).

  1. In Line 153-156,Please elaborate on the experimental methods and procedures in this section.

This assay is described in detail in Reference 48, but to satisfy the Reviewer, details of the MT assay are now expanded in the revised manuscript (lines 157-164) in Section 2.9.

Results 

  1. In Line 173,please elaborate on the significance of MmΔIBBIMPα protein construction regarding the topic of this article. 

The MmΔIBBIMPα protein construction has been described previously in detail (References 14, 16-18) as well as in lines 75-77 of the original submission. It is a form of IMPa that is not autoinhibited and hence can bind NLSs with high affinity in the absence of IMPb1. We now spell this out clearly in the results section (lines 184-186) with new Reference 52 as appropriate, and thank the Reviewer.

  1. Figure 1,the chemical structures of the two drugs are presented in the article.  At what specific sites are they targeting the IMP protein? 

The drugs target IMPα within the ARM repeat domain, but detailed information would require X-Ray crystallographic analysis of a complex of IMPα with the respective compounds – this is obviously outside the scope of the current study, especially as no crystal structure is yet available for apicomplexan IMPα.

  1. Figure 4,What is the purpose of the interaction between moxidectin and ivermectin with MmIMPα?

See response to Major comment 3; moxidectin binding to importin alpha unfolds the protein, thereby inhibiting its ability to bind NLSs/IMPbeta1 (see Discussion lines 307-318).

  1. What is the homology between Plasmodium IMP and host-associated protein, and does Moxidectin have an impact on the host?

We have added the homology information to the text (Lines 182-183). As indicated above and in the manuscript, moxidectin is FDA-approved for human use in 2018 – no reports of side effects exist thus far (see References 36-40; 49, 50, 54). 

  1. Table 4,Ivermectin and Moxetine exhibited high cytotoxicity towards HFF cells.  What was the aim of this experiment, or why were other relevant cytotoxicity experiments not carried out? 

The aim of the experiments in Table 4 was to assess moxidectin activity in an infectious assay with T. gondii that uses primary human foreskin fibroblasts (HFF cells) as the host. We have spelled this out more clearly (lines 294-295; and see following text) – we thank the Reviewer.

  1. The picture definition is excessively low. Please make modifications. 

Figures have been submitted as high resolution tiff/jpg files. We thank the Reviewer.

Discussion:

  1. In Line 303-307,the discussion section is not in line with the research. Please make the necessary revisions.

Revisions to lines 303-307 have been made – we thank the Reviewer. Importantly, we have added a clearer summary/conclusion of the results of the study (lines 307-318) – we thank the Reviewer.

  1. In Line 318-323 “low µM concentration” this description is incorrect. Please amend it.

We have amended the text – we thank the Reviewer.

  1. There is no summary of the main results in the discussion section.  Please add that the discussion should center on the main experimental results.

As indicated above, we have added a summary of the main results at the end of the first paragraph of the Discussion as requested (lines 307-318).

We thank the Reviewer for the positive and probing comments, which have been instrumental to making the manuscript a better and more rigorous evocation of our results.

Reviewer 2 Report

Comments and Suggestions for Authors

The authors report the action of an ivermectin analogue, moxidectin, against apicomplexa parasites. The experimental design and execution is appropriated and well performed. My major concern is the lack of novelty regarding the high degree of similarity between both compounds. I suggest authors to at least compare ivermectin and moxidectin binding importin alpha/beta heterodimer to gain more insights into the mechanism of action.

Figures are also very poor and need to be improved. 

Author Response

We thank the Reviewer for the positive, constructive comments. Our responses are below:

The authors report the action of an ivermectin analogue, moxidectin, against apicomplexa parasites. The experimental design and execution is appropriated and well performed.

We thank the reviewer for the positive comments.

My major concern is the lack of novelty regarding the high degree of similarity between both compounds.

As explained in the manuscript, the novelty is that the implication of the results is that there is a class effect, and that various different avermectins/milbemycins may be candidate molecules targeting importins and hence potential antimicrobials – this point has now been strengthened in the manuscript, particularly in the first paragraph of the Discussion (lines 306-323).

I suggest authors to at least compare ivermectin and moxidectin binding importin alpha/beta heterodimer to gain more insights into the mechanism of action.

Perhaps the Reviewer has not appreciated it, but Figure 2 addresses this precise point, and supports the idea that both inhibitors have very similar MOAs. For the Reviewer’s information, no apicomplexan Importin beta has yet been analysed, which is why we tested mammalian importin beta1 - we have added this information to the manuscript (line 208-212). We thank the Reviewer.

Figures are also very poor and need to be improved. 

Figures are all provided at high resolution as tiff/jpeg files.

We thank the Reviewer for the positive, constructive comments, which have been instrumental to making the manuscript a better and more rigorous evocation of our results.

Reviewer 3 Report

Comments and Suggestions for Authors

The development of novel therapeutic strategies is crucial to combat resistant malaria parasites. In this context, Walunj et al. reported that the ivermectin-related compound, moxidectin, exhibited inhibitory effects on Plasmodium falciparum and Toxoplasma gondii in vitro. While the data is promising, several improvements are required before the study is ready for publication:

  1. The authors state that moxidectin and ivermectin target apicomplexan glutamate-gated chloride channels and demonstrate binding to importin α. However, the structural basis for these interactions remains unclear. A molecular docking study should be included to elucidate the potential mechanisms of action.
  2. Understanding the genetic diversity of drug targets is essential for assessing drug efficacy and resistance development. The study should provide or discuss polymorphism data for importin α. Additionally, the importin α gene ID in PlasmoDB should be included to facilitate follow-up studies—likely Pf3D7_0812400?
  3. In Figure 1, labels for panels A and B are missing, and the legend for panel A is incomplete.
  4. In section 3.1, the data presented in the text, table, and Figure 1 are inconsistent and need to be reconciled.
  5. In Table 2, the protein names (ΔIBBmIMPα or MmIMPα) are unclear. Furthermore, the values in this table do not align with the data shown in Figure 4.
  6. In section 3.3, the data in Figure 5 are inconsistent with the corresponding table 3 and need correction.

Author Response

We thank the Reviewer for the positive, constructive comments. Our responses are below:

The development of novel therapeutic strategies is crucial to combat resistant malaria parasites. In this context, Walunj et al. reported that the ivermectin-related compound, moxidectin, exhibited inhibitory effects on Plasmodium falciparum and Toxoplasma gondii in vitro. While the data is promising, several improvements are required before the study is ready for publication:

  1. The authors state that moxidectin and ivermectin target apicomplexan glutamate-gated chloride channels and demonstrate binding to importin α. However, the structural basis for these interactions remains unclear. A molecular docking study should be included to elucidate the potential mechanisms of action.

We thank the Reviewer for pointing out our oversight – we meant to say that it is known that the chloride channels of helminths and insects are targeted (analysis of apicomplexan glutamate-gated chloride channels has not been performed) – we now clarify this point (lines 324-325), including several new References (new References 54-58). We thank the Reviewer for highlighting our mistake.

  1. Understanding the genetic diversity of drug targets is essential for assessing drug efficacy and resistance development. The study should provide or discuss polymorphism data for importin α. Additionally, the importin α gene ID in PlasmoDB should be included to facilitate follow-up studies—likely Pf3D7_0812400?

We thank the Reviewer for highlighting this important issue – we now make it clear that mammalian systems have multiple importin alpha forms, whereas P. falciparum and T. gondii only have one (lines 181-184, with homology information importantly now included) – we have also included the gene IDs for all of the proteins used in the study, as appropriate (lines 76-79; indeed the P. falciparum importin α gene ID is Pf3D7).

In Figure 1, labels for panels A and B are missing, and the legend for panel A is incomplete.

We have amended the figure and the legend – we thank the Reviewer for pointing this out.

  1. In section 3.1, the data presented in the text, table, and Figure 1 are inconsistent and need to be reconciled.

There is no inconsistency – Figure 1/2 each embody one of the three measurements encapsulated in the mean values shown in Table 1 (as spelled out in the footnote to the table – line 230). This is also spelled out in the respective Figure legends (lines 207-208; 225-227).

  1. In Table 2, the protein names (ΔIBBmIMPα or MmIMPα) are unclear. Furthermore, the values in this table do not align with the data shown in Figure 4.

We thank the Reviewer for highlighting the fact that there are a number of instances in the text where MmΔIBBIMPα is written incorrectly as ΔIBBmIMPα; we have now amended these throughout, including in Table 2.

  1. In section 3.3, the data in Figure 5 are inconsistent with the corresponding table 3 and need correction. 

There is no inconsistency - Figure 5 embodies one of the three measurements encapsulated in the mean values shown in Table 3 (as spelled out in the footnote to the table – line 293). This is also spelled out in the Figure 5 legend (lines 289-290).

We thank the Reviewer for the positive, constructive comments, which have been instrumental to making the manuscript a better and more rigorous evocation of our results.

Round 2

Reviewer 1 Report

Comments and Suggestions for Authors

It's ok.

Reviewer 2 Report

Comments and Suggestions for Authors

Authors improved the manuscript.

Reviewer 3 Report

Comments and Suggestions for Authors

I am satisfied with the revised manuscript, which has addressed all my major comments. I fully support its publication.